# Impact of Urbanization on Pluvial Flooding: Insights from a Fast Growing Megacity, Dhaka

Md Shadman Sakib, Siam Alam, Shampa *, Sonia Binte Murshed, Ripan Kirtunia, M. Shahjahan Mondal and Ahmed Ishtiaque Amin Chowdhury



Institute of Water and Flood Management (IWFM), Bangladesh University of Engineering and Technology (BUET); Dhaka 1000, Bangladesh; shadman@iwfm.buet.ac.bd (M.S.S.); siam94015@gmail.com (S.A.); sonia@iwfm.buet.ac.bd (S.B.M.); ripankirtuniajust@gmail.com (R.K.); mshahjahanmondal@iwfm.buet.ac.bd (M.S.M.); ishtiaquechowdhury@iwfm.buet.ac.bd (A.I.A.C.)
* Correspondence: shampa_iwfm@iwfm.buet.ac.bd

**Abstract:** The 400-year history of Dhaka says that the city once had several well-known natural canals (khals) that drained stormwater and graywater. In addition to city's combined sewer system, these water bodies offered an essential natural drainage system that allowed to manage the monsoon rainfall effectively. However, over the past three decades, due to rapid urbanization, these khals have significantly depleted to the point where they are no longer capable of draining the city's monsoon runoff. Using past, present, and future Land Use and Land Cover (LULC) and urban drainage modeling, this study identified the effects of such LULC change on pluvial flooding of the northern part of the city. Analysis shows that the rapid and extensive changes in LULC over the past decades have resulted in significant shrinkage of these khals, consequently leading to escalated rates of urban flooding in this region. The western part of Turag thana, low-lying areas close to the Baunia Khal depression, and the upstream region of Abdullahpur Khal are highly vulnerable to future urban floods. The projected LULC change indicates an increase of 8.47%, 8.11%, and 4.05% in the total inundation area by 2042 for rainfall events with return periods of 50 years, 25 years, and 2.33 years, respectively. The findings also indicate that 11% more area is likely to experience long-duration flooding due to LULC change.

**Keywords:** LULC; pluvial flooding; SWMM; MOLUCSE; Dhaka city

## 1. Introduction

Pluvial floods in megacities are a serious issue worldwide, particularly for rapidly growing cities [1–4]. Rapid urban development in these megacities has led to the unchecked conversion of surface water bodies, greeneries, and buffer zones to paved urban land, thereby increasing its susceptibility to flooding [5]. It also alters the hydrological balance of the city's intricate canal-stream network which forms the backbone of the natural drainage system [6]. However, this is not the only prime factor that contributes to the waterlogging problem, rather variables including heavy seasonal precipitation, rapid shift in built-up land use patterns, inadequate drainage facilities, and ill waste management practice all play critical roles [7–11]. Over the past four decades, South Asian countries have shown a characteristic trend of unplanned urban expansion, which leads to water insecurities, environmental degradation, public health, and sanitation issues [12]. By 2050, it is expected that approximately 70% population of the world will have migrated to cities, with Asian and African continents accounting for 93% of the migration [13]. This demographic shift adds to the already depleted environmental resource (green space 5%), particularly in densely populated South Asian capitals such as Dhaka, Kathmandu, and Karachi, and is expected to exaggerate the problem [14]. This paper focuses on such a problem in one of the South Asian capitals-Dhaka in Bangladesh, home to over 22 million as of 2022 [15].

The 400-year history of Dhaka shows that it was a low-lying floodplain of multiple river systems [16].

It is surrounded by the Turag-Buriganga river system to the west and the Balu-Shitalakhya rivers to the east [17]. Records show that, in the latter half of the 19th century, Dhaka was subjected to severe fluvial flood events at regular intervals in 1954, 1955, 1962, 1966, 1974, 1987, 1988, and 1998. Among these 1988 flood was the most devastating. It was triggered by excessive fluvial inflow from the surrounding rivers [18]. The flood peak lasted for three weeks, inundating 85% of the city and setting record-high water levels [19]. In response, the government initiated the 'Dhaka City Integrated Flood Protection Project (DCIFPP)' in 1991 to protect existing and future development in low-lying built-up areas [20]. The project oversaw the construction of 30.2 km of dyke, 9.25 km of flood wall, 12 numbers of floodproofing sluices, and the installation of 3 pumping stations at Dholai Khal, Kalyanpur, and Goranchatbari. Before the construction of the embankment following the flood in 1991, the city experienced several instances of pluvial flooding caused by the Turag-Buriganga river system. Following the implementation of DCIFPP in 1991, the city was safeguarded against flood occurrences. Therefore, currently, it is not prone to pluvial floods.

However, rainwater that once flowed into the river was now trapped within embanked areas, resulting in localized flooding. This was due to poor maintenance of the embankment's flow control structures, blocked sluice gates from solid waste dumping, and rampant urbanization which collectively disrupted Dhaka's core drainage network [21]. As a consequence, in 1998, Dhaka suffered its worst pluvial flood resulting in very high rainfall. Most of eastern Dhaka was inundated, along with 20% of the western part, which was newly embanked under the flood action plan [19]. Over time, the majority of the sluice gates became inoperable. Despite flood embankment in the west, the flooding problem had not been solved; rather, the inundation dynamics had shifted from fluvial to pluvial floods.

Pluvial flood dynamics are linked to urbanization, which alters the natural land surface through impervious artificial structures such as asphalt, concrete, brick, stone, and buildings. This reduces the amount of rainfall-runoff that can infiltrate into the substrata, resulting in flooding. Especially in the case of rapidly growing megacities like Dhaka, it is pivotal to understand its Land Use and Land Cover (LULC) dynamics. Denser inner-city areas with high runoff potential have nearly tripled in the last four decades [22]. A survey by [23] shows the primary arterial canals in Dhaka to be infringed up to 40% of their original demarcation. In addition, the widespread illegal encroachment of waterways and ponds (small waterbodies) has reduced the amount of open space available for water to flow and be stored [24]. The problem is aggravated by poor waste management practices and rapid population growth which have put pressure on the already inadequate drainage infrastructure [11]. These factors collectively reduce the city's ability to manage the sudden influx of runoff following moderate to heavy rainfall, leading to severe pluvial flooding.

As previously stated, the three pump stations, Kallyanpur, Goranchatbari, and Dholai Khal, act as flood control structures to protect Dhaka. However, the retention ponds at Kallyanpur and Dholai Khal have been significantly reduced in recent decades due to aggressive encroachment, landfilling, and residential development [25].

This advertently reduces the runoff accumulation lead time, resulting in severe flooding under medium to heavy rainfall. The only exception is the Goranchatbari Pump Station (GPS), which still has a functioning retention pond of 676 acres [26]. It is responsible for draining out stormwater from the north-western part of greater Dhaka (Figure 1), i.e., Mirpur, Pallabi, Cantonment, Uttara, Diabari, and Tongi. Nevertheless, GPS is experiencing immense pressure due to aggressive urbanization, which has compelled it to give up a considerable portion of its retention pond for urban settlements and residential expansion [27]. Therefore, it is critical to understand and predict the system's response to this disruptive LULC trajectory and understand its potential impact on escalating urban flood vulnerability.

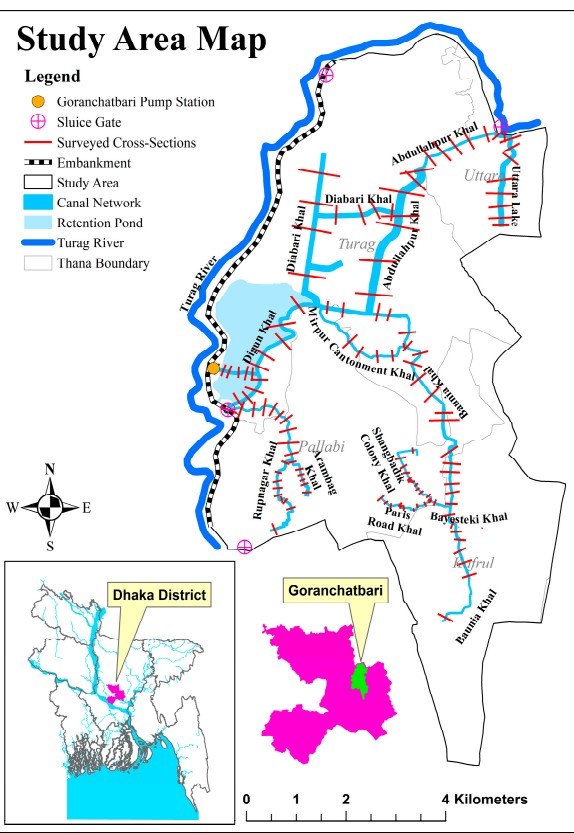

**Figure 1.** Map of Goranchatbari catchment for drainage study.

The flood dangers in Dhaka were evaluated by [28] using the utilization of geographic information systems (GIS) and remote sensing (RS) imagery. This assessment involved the analysis of flood frequency and flood depth data. Based on the flood extent maps spanning the years 1988 to 2009, a frequency map was generated to illustrate the areas affected by floods. The analysis revealed that a minimum of 23% of the total area fell inside the high-flooded zone. On the other hand, the flood depth map indicated that a significant portion, specifically 45%, of the surveyed region was characterized by a substantial level of flooding. That study proposes to create a comprehensive rainfall-runoff model that incorporates hourly or daily time stamps in order to accurately capture the dynamics of floods and their seasonal variations. In their study, Ref. [29] utilized an integrated hydrologic and hydrodynamic model to investigate the phenomenon of urban flooding in Dhaka City. However, it is important to note that their research primarily concentrated on the specific area of Segunbagicha kahl. [25] conducted an assessment of the efficacy of several flood control measures implemented in the vicinity of Dhaka city. Their objective was to identify the factors contributing to both external and internal flooding of the protected areas in Dhaka during significant flood events that occurred in the recent past. Their primary emphasis was on fluvial flooding only. In their study, Ref. [30] conducted an assessment of long-term urban surface water changes utilizing multi-year satellite imagery. Their findings revealed the occurrence and transitions in surface water are significantly influenced by the nature of the city's expansion, seasonal variations, and geographic locations. However, the study did not take into account the dynamics of flooding and future land use. Ref. [31] focus on the identification of areas prone to water logging risks in Dhaka city and the assessment of their flood susceptibility. The flood susceptibility map was generated in order to assess the potential for waterlogging. This study also lacks the future projection of land use.

Under these backdrops, this research investigates the pattern of LULC change in the Goranchatbari sub-catchment and then identifies its potential impact on pluvial flooding.

Specifically, we attempt to determine the pattern of change in LULC over the last five decades; forecast the future LULC and finally examine how the pluvial flooding dynamics are changing due to altered LULC using historical precipitation data and one-dimensional (1D) urban drainage modeling.

*Study Area*

Dhaka, the capital of Bangladesh, is located between 23°40′ N to 23°54′ N latitude and 90°02′ E to 90°31′ E longitude. It is the most densely populated megacity in the world with an area of approximately 258.78 km$^2$. Its ground elevation varies between 1.5 and 15 m above Mean Sea Level (MSL), with an average elevation of 6 m [32]. The city is divided into 41 administrative sub-divisions called 'Thana', which are managed by two independent organizations, i.e., Dhaka South and Dhaka North City Corporations (DSCC and DNCC). The greater Dhaka is divided into 10 major artificial catchments for stormwater and drainage management [33]. Among these Goranchatbari catchment stands to be the largest under DNCC covering an estimated area of 67.52 km$^2$. It stands out from other drainage catchments in Dhaka city due to its environmental diversity. The catchment still retains most of its natural canal pathways, despite aggressive urban development. In addition, the presence of a comparatively larger amount of vegetation or greeneries and waterbodies has been one of the prime reasons behind selecting it as the study area [34,34]. However, just like other parts of Dhaka, the waterbodies of this region are also facing severe anthropogenic encroachments over the last decade [35].

The catchment (Figure 1) is drained by a 66 m$^3$/s capacity pumphouse located at Goranchatbari. Its 676-acre ponding area stores stormwater runoff from Cantonment, Turag, Kafrul, Mirpur, Shah Ali, Pallabi, and Uttara which is pumped out into the Turag River. The study area consists of 11 major natural canals (Table 1) having a total length of 37.06 km. Although originally designed to serve as stormwater channels, these are subjected to additional loads to carry industrial and domestic wastewater. Consequently, these canals suffer from severe solid waste accumulation, and blockage and fail to function at their design capacity [36]. The detailed status of several khals is discussed in Supplementary S1.

**Table 1.** Average width, length, and drainage area of major natural canals within the Goranchatbari catchment.

| SL | Khal Name | Average Width (m) | Length (km) | Contributing Thana |
|----|-----------|-------------------|-------------|---------------------|
| 1 | Baunia Khal | 22.50 | 9.61 | Cantonment, Turag, Kafrul, Mirpur |
| 2 | Diabari Khal | 81.28 | 6.89 | Turag |
| 3 | Digun Khal | 34.11 | 4.33 | Turag |
| 4 | Eastern Housing Khal | 14.65 | 4.27 | Shah Ali, Mirpur, Pallabi |
| 5 | Mirpur Cantonment Khal | 16.94 | 3.30 | Pallabi, Turag |
| 6 | Abdullahpur Khal | 74.84 | 2.64 | Uttara, Turag |
| 7 | Uttara Lake Khal | 58.46 | 2.34 | Uttara |
| 8 | Shangbadik Colony Khal | 6.32 | 1.27 | Pallabi |
| 9 | Paris Road Khal | 6.61 | 1.14 | Pallabi |
| 10 | Arambag Khal | 12.80 | 0.87 | Pallabi, Shah Ali |
| 11 | Bayeshteki Khal | 10.96 | 0.41 | Pallabi, Kafrul |

The present area of the Goranchatbari detention pond measures 636.5 acres. It serves as a reservoir for waste and stormwater originating from the localities of Mirpur, Cantonment, Airport, Pallabi, Shah Ali, and Uttara thana. In the dry season, the amassed wastewater is discharged via gravitational force into the Turag River by means of a drainage sluice positioned near the pump house. However, during the rainy season, it becomes necessary to employ pumping mechanisms to transfer the excess stormwater into the Turag River, as the river's water level is significantly higher. The GPS comprises three pump houses, each containing three vertical turbine pumps with an individual pumping capacity of 7.33 m$^3$/s.

However, at the time of the study, one of the pumps in pump house-1 was undergoing maintenance, resulting in a reduction of the current pumping capacity to 58.64 m$^3$/s. In the monsoon season, the pumps are operated to maintain a water level of 3.5 m in the retention pond. This is achieved by running one or two pumps in a periodic manner over the course of several days following a heavy precipitation event.

Figure 2a shows pump house 1's intake channel, while Figure 2b illustrates the on-site solid waste disposal site situated adjacent to the intake channel within the premise of the pump station. Various forms of household waste, such as plastic bags, used furniture, and dead tree branches, along with inorganic materials, enter the pumping channel and necessitate manual screening to prevent pump clogging. Figure 2c provides a view of the 636.5-acre retention basin of the pump house, which has undergone a 50-acre reduction due to the development of a newly constructed residential area visible in the far right of the image. This expansion took place after 2016 to meet the residential demands of the city by land reclamation from the retention basin.

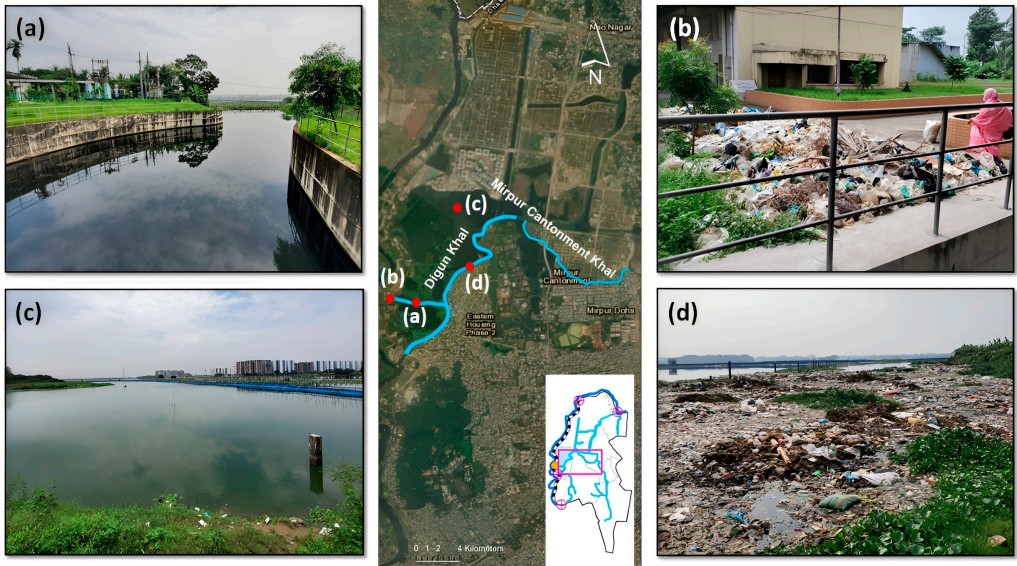

**Figure 2.** Field Observation—GPS and Digun Khal: (**a**) pump house-1 intake channel at GPS (**b**) solid waste disposal site adjacent to the pump house-2 at GPS; (**c**) GPS retention pond and RAJUK Uttara Apartment Project residential area in the upper right quadrant (**d**) Solid waste and floating debris accumulation at the periphery of the GPS retention basin.

The average bed elevation of the retention basin ranges from −0.5 to −1.0 m PWD. Additionally, an artificially excavated canal with a bed level of −1.5 m runs perpendicular to the pump station's intake openings, facilitating the transfer of retained water from the reservoir to the pump intake channels. Figure 2d exhibits an image near the inlet of Digun Khal, which connects the Cantonment and Eastern Housing Area. Notably, a substantial amount of floating waste and debris can be observed lining the periphery of the khal. During the time of the study, the water in these areas was stagnant and exhibited biohazardous quality. The detailed status of the khals in the study area is presented in Supplementary Section S1.

## 2. Materials and Methods

The research is divided into two parts: detecting changes in LULC and assessing the impact of changed LULC on pluvial flooding using 1D modeling. Satellite image analysis revealed the evolution of LULC over the last five decades. The spatial variable (i.e., Waterbodies, Green areas or builtup areas) for LULC was identified while analyzing the LULC, and future land cover was predicted using these variables. From the last 66 years of measured precipitation, 2.33-, 25-, and 50-year return period rains were calculated, and

these return period rains were used to calculate the impact of pluvial flooding on land use change. Statistically, the 2.33-year return period flood is considered as an average flood. Therefore, the 2.33-year storm event was chosen as the reference storm to establish a control case. It also aligns with the return period of the mean annual flood, which is also 2.33 years. Figure 3 shows the overall methodology of the study.

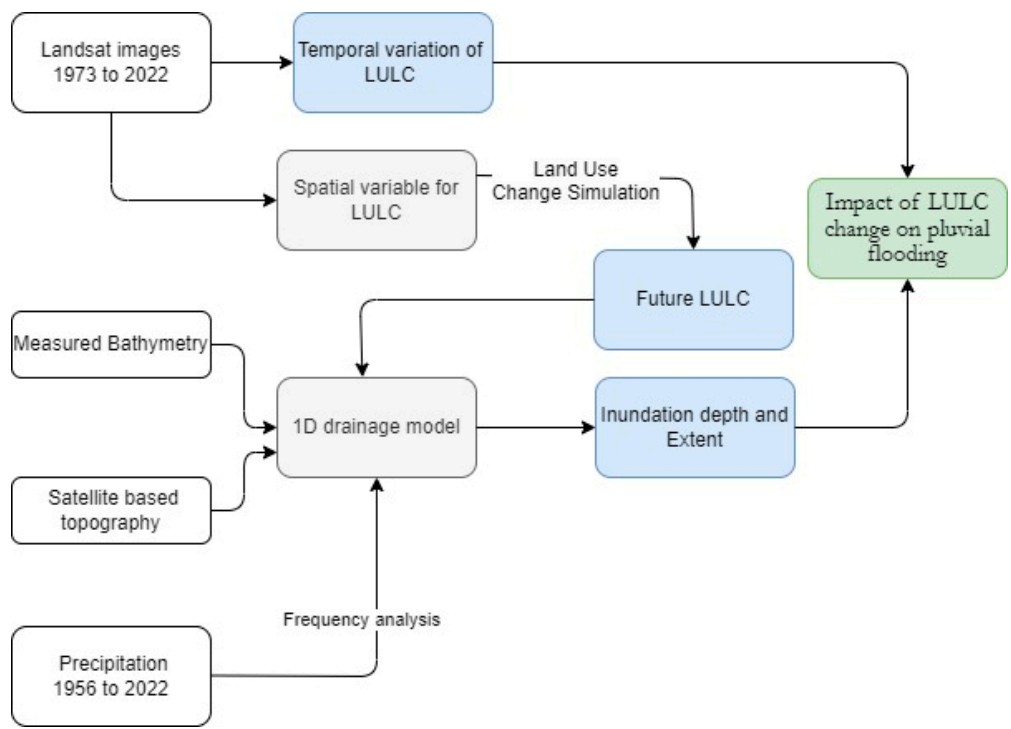

**Figure 3.** The overall methodology of the study.

*2.1. Assessment of LULC*

The analysis of Landsat satellite images from the years 1973, 1983, 1993, 2003, 2013, and 2022 is conducted to detect changes in LULC. The RGB composite images were subjected to classification using a clustering algorithm that relied on their spectral features. Each year, the iteration commenced with a total of thirty courses and concluded with three classes representing waterbodies, green areas, and built-up areas. After each processing iteration, the clusters that were formed underwent labeling based on the class to which they might be assigned. This labeling procedure was conducted using image processing capabilities available in ArcGIS. The methodology of LULC assessment is shown in Figure 4. Accuracy assessment was performed using the Kappa statistic by [37]. As the reference dataset, Google Earth Pro images were used to determine the accuracy of the classified result. Randomly 60 points were generated in several LULC types and were compared with the reference dataset. The Kappa coefficient was 0.85 which indicates a good agreement according to [37]. This assessment was done for the land use of the year 2022. This is due to the fact that the future LULC was validated using the LULC of 2022. The confusion matrix is shown in Table 2.

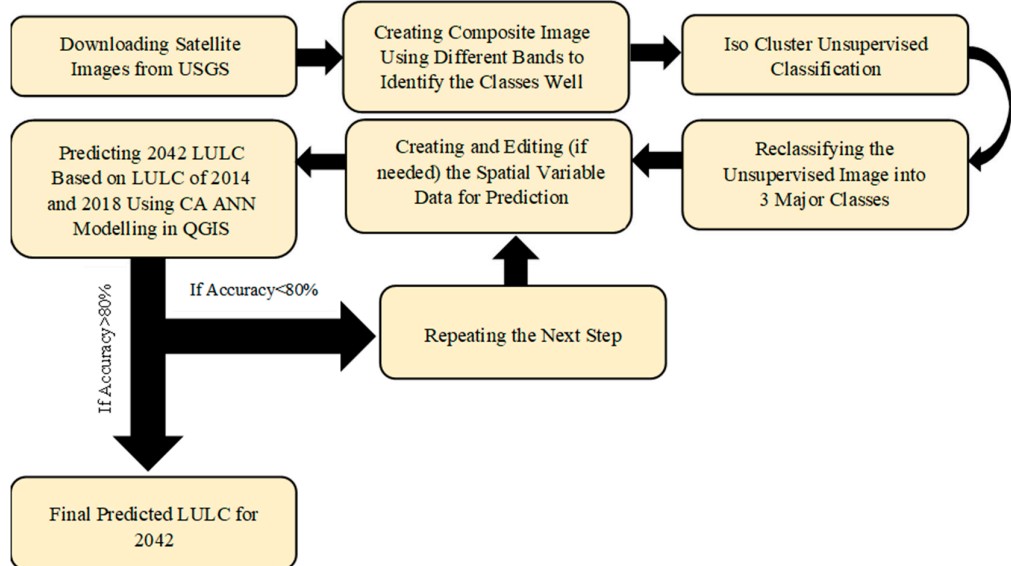

**Figure 4.** Overall Workflow of the Landuse Analysis.

**Table 2.** Accuracy assessment of LULC.

| Class Value | Waterbodies | Green Area | Built-Up Area | User Accuracy |
|---|---|---|---|---|
| Waterbodies | 15.00 | 2.00 | 0.00 | 0.88 |
| Green Area | 0.00 | 15.00 | 1.00 | 0.94 |
| Built-up Area | 0.00 | 2.00 | 15.00 | 0.88 |
| Producer Accuracy | 1.00 | 0.79 | 0.94 | 0.90 |
| Kappa | 0.85 | | | |

## 2.2. LULC Prediction

This study applied a hybrid Cellular Automata (CA)—Artificial Neural Network (ANN) based Multi-Objective Land Use Change Simulation Environment (MOLUCSE) model [38–42] in QGIS to predict the future LULC change. Similar mathematical models e.g., Markov Chain Model (MCM) [43,44], Cellular Automata (CA) [45,46], Artificial Neural Network (ANN) [38,47], and advanced remote sensing techniques [22,48] have been applied to predict the LULC changes in other studies. Landsat satellite imagery for 1973, 1980, 1993, 2003, 2013, and 2022 is analyzed for LULC change using the ArcGIS supervised classification [49]. The elements of the LULC are clustered based on their spectral characteristics. Following that, pixels were grouped based on their spectral values, and noise was filtered out using image processing tools in ArcGIS. The classification was verified using the field observations around the retention pond of Goranchatbari pumphouse, low-lying areas of Baunia khal, and highly built-up areas of Diabari, Uttara, Pallabi, Mirpur, and Vashantech. Subsequently, the images were reclassified into three categories, i.e., waterbodies and canals, vegetation, and settlements plus bare lands [50]. These data were fed into MOLUCSE along with a digital elevation model, population density, and distances from roads and waterbodies raster [46,51] to predict the future LULC pattern for the Goranchatbari catchment. These spatial input variable for LULC prediction over the Goranchatbari catchment area is shown in Figure 5. Whereas, Figure 6 shows the workflow on which the prediction of future land use was prepared using CA-ANN algorithm in QGIS. Number of hidden layers used in this step was 10. The rest of the parameters like neighborhood, number of samples, etc. were prepared through a trial-and-error method to find the best possible case in training the model. The year 2042 was chosen to assess the drainage condition immediately after achieving the national goal "Vision 2041". The Bangladesh Government has a plan to achieve High Income Country status by 2041 which

is why the prediction period for the land use was set at 2042 to incorporate the possible negative sides of wealth and urbanization [52,53].

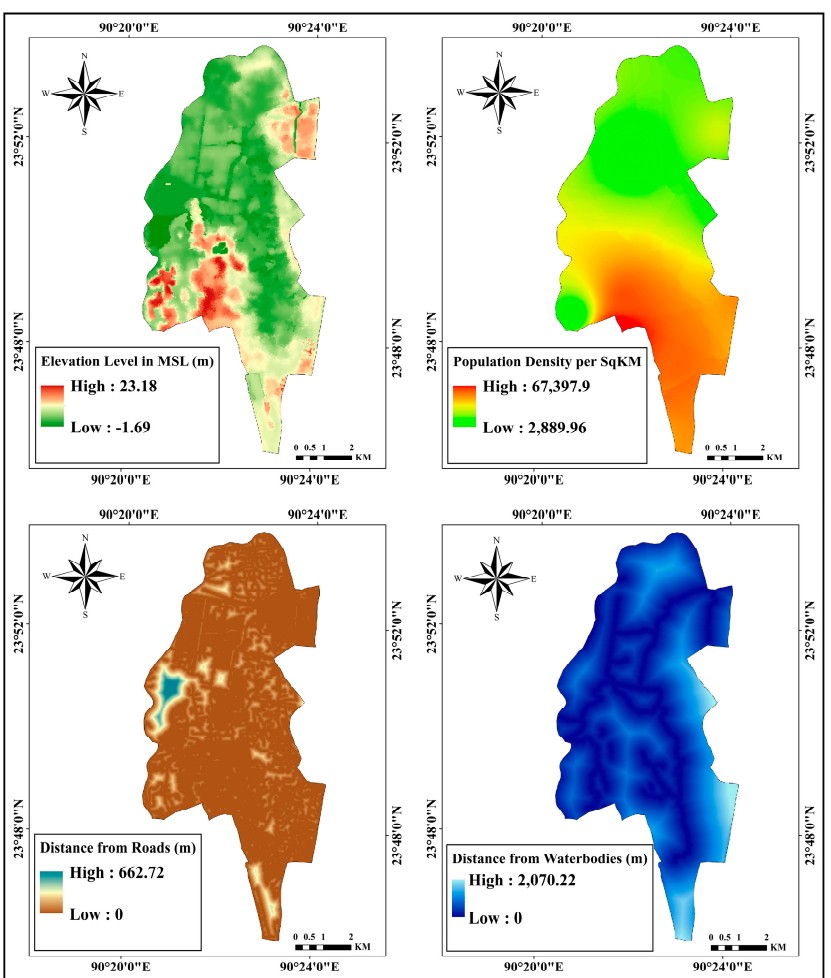

**Figure 5.** Spatial variable for LULC prediction using MOLUSCE plugin in QGIS.

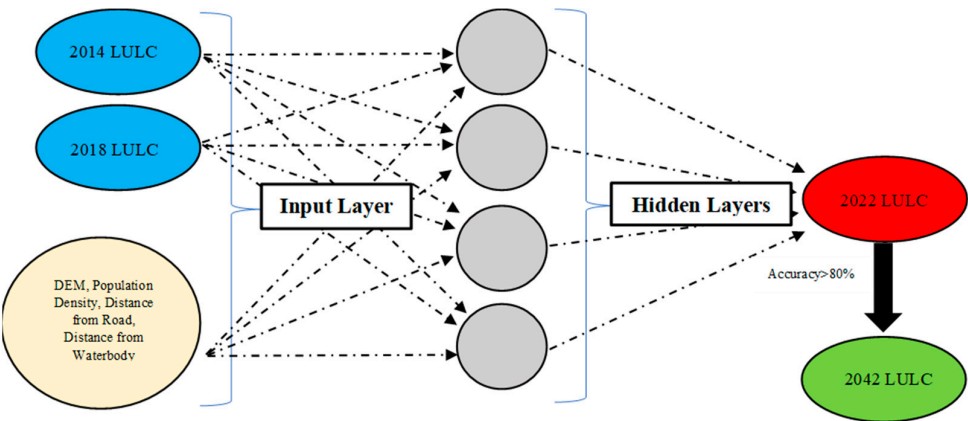

**Figure 6.** CA ANN Model Algorithm Work Flow.

Kappa Coefficient (κ) was the determinant factor in assessing the accuracy of the prediction [54,55]. The model parameters are set to have a high κ value. According to [56,57] a model is considered to be of high accuracy for prediction if its kappa value is over 0.8, while a value in the range of 0.61 to 0.8 is seen as being acceptable for use in making future predictions. The observed and predicted LULC using MOLUSE is shown in

Figure 7. for which the validation kappa was calculated to be 0.625. The prediction was found to be representative with an overall accuracy of 86.46%.

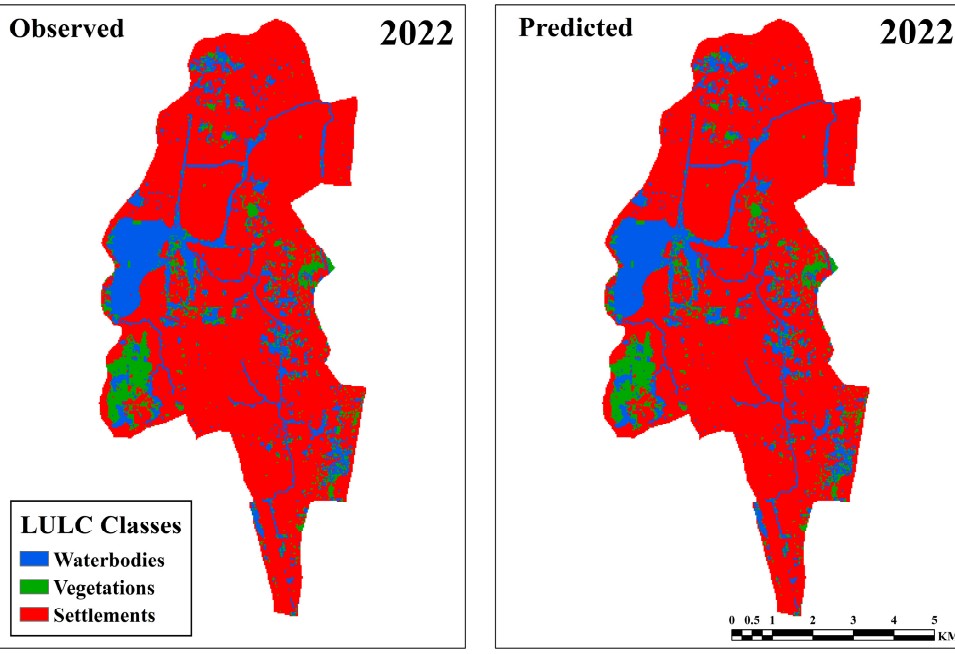

**Figure 7.** LULC classification validation between observed Landsat data (2022) and MOLUSCE-based simulated case (2022).

*2.3. Urban Drainage Modeling and its Validation*

Governing Equations

In this study, the Personal Computer Storm Water Management Modeling (PCSWMM) software (Version 7.4.3240) is used to model the urban drainage system of the Goranchatbari catchment. The hydrodynamic component of the PCSWMM model is based on a one-dimensional Saint Venant momentum equation [58] for simulating gradually varied and unsteady flow conditions. Manning's equation [59] was applied to assess the hydraulic characteristics of the conduit flow, with Manning's "*n*" value serving as the primary parameter for calibration. To model the infiltration of rainfall, the Green-Ampt model is employed, while for flood routing, a dynamic routing algorithm proposed by [60] is utilized. These are part of dynamic wave analysis in EXTRAN [61]. The next paragraphs demonstrate the governing equations used in the model.

$$\frac{\partial d}{\partial t} = r + i - e - f - Q \tag{1}$$

Equation (1) is the hydrologic governing equation. It is derived from the Conservation of Mass Law [62]. Here, the change of depth per unit of time in sub-catchments is expressed in the form of $r$, $i$, $e$, $f$, and $q$. $r$ is the rate of upstream run-on, $i$ is the intensity of rainfall, $e$ is the rate of evaporation, $f$ is the rate of infiltration, and $q$ is the rate of runoff. All of them can be expressed as flow rates per unit area (m/s). In the 1D model each sub-catchment is treated as a nonlinear reservoir. It is assumed that the capacity of this reservoir is the maximum when the topographic depression is higher. Surface runoff $Q$, occurs when the depth of water in the sub-catchment exceeds the maximum depression storage. The depth of water over the sub-catchment is updated in real time by numerically solving a water balance equation over the sub-catchment.

$$\frac{\partial A}{\partial t} + \frac{\partial Q}{\partial x} = 0 \tag{2}$$

Equation (2) is the continuity equation where $x$ is referred to as distance, $t$ is time, $A$ is the flowing cross-sectional area, and $Q$ is the flow rate. It is also the hydraulic governing equation of PCSWMM.

$$\frac{\partial Q}{\partial t} + \frac{\partial\left(\frac{Q^2}{A}\right)}{\partial x} + gA\frac{\partial H}{\partial x} + gAS_f + gAh_l = 0 \tag{3}$$

Equation (3) is the 1D Saint Venant momentum equation. Here, $H$ is the hydraulic head of water in the conduit which is the summation of invert elevation and water depth, and $S_f$ is the friction slope. Manning's equation is used to determine the flow inside the conduits where Manning's coefficient represents the roughness parameter, as shown in Equation (4). Here, $Q$ is the flow in the conduit, the cross-sectional area of the conduit is marked as $A$, the hydraulic radius is denoted as $R$, and $S$ is the slope of the ends of the conduits. The roughness coefficient, $n$ is used as a calibration parameter.

$$Q = \frac{1.49}{n} A R^{\frac{2}{3}} S^{\frac{1}{2}} \tag{4}$$

$h_l$ is the local energy loss per unit length of conduit. In the case of solving a network of conduits, an additional continuity relationship is needed for the junction nodes that connect two or more conduits together. To solve this a continuous water surface is assumed to exist between the water elevation at the node and in the conduits that enter and leave the node. The change in hydraulic head $H$ at the node with respect to time was calculated using Equation (5).

$$\frac{\partial H}{\partial t} = \frac{\sum Q}{A_n + \sum A_c} \tag{5}$$

Here where $A_n$ is the surface area of the node, $\sum A_c$ is the surface area of the conduits connected to the node, and $\sum Q$ is the net flow into the node contributed by all conduits connected to the node as well as any externally imposed inflows. Dynamic Wave routing was utilized because it couples the solution for both water levels at nodes and flow in conduits. For calculation of infiltration Green-Ampt Infiltration Method has been implicated in this study [63]. This method for modeling infiltration assumes that a sharp wetting front exists in the soil column, separating soil with some initial moisture content below from saturated soil above. Equation (6) shows the basic equation of the Green-Ampt method. Here, $f$ is the rate of infiltration which is expressed in terms of $K$, $\varnothing$, $\theta_i$, $\psi$, and $F$. $K$ is the hydraulic conductivity, $\varnothing$ is the porosity, $\theta_i$ is the initial water content, $\psi$ is the wetting front soil suction head, and $F$ is the cumulative infiltration.

$$f = K\left[1 + \frac{(\varnothing - \theta_i)\psi}{F}\right] \tag{6}$$

To model pumps that connect two nodes together, the flow is calculated based on the heads at their end nodes.

Model Schematization

The topography of the urban drainage model is based on the Copernicus Digital Elevation Model (COP-30 DEM). It provides elevation data with a spatial resolution of 30 m. The data source for precipitation, LULC analysis, and projection is listed in Table 3. A professional survey team equipped with echo sounder and survey equipment was assigned to measure canal depth and cross-section. This survey conducted in this study consisted of a total of 126 cross-sections, which were distributed among 11 major canals listed in Table 1 and whose location is shown in Figure 1. The distribution of surveyed points within each canal was determined based on canal length, alignment, and variations in width. This was guided by engineering judgement and was further substantiated by multiple field visits. The spacing between surveyed points ranged from 40 m to 900 m on an average of 300 m. But it is crucial to note that our survey strategy was designed to ensure comprehensive

coverage of all significant features of the canals, including abrupt constrictions and width variations. PCSWMM hydraulic model does not facilitate interpolation between channel cross-sections, therefore, a uniform open channel cross-section was considered between the surveyed points.

**Table 3.** Input data used and their sources for urban drainage modeling.

| Data Type | Data Source | Function |
|---|---|---|
| Daily Rainfall data (1965–2020) | Bangladesh Meteorological Department (BMD) | Frequency analysis |
| 3 Hourly Rainfall (2020) | Bangladesh Meteorological Department (BMD) | Model validation |
| Sentinel-2 Image | NASA Earth Data | Model input as percent impervious |
| Landsat 4, 5, 8, 9 Image | USGS Earth Explorer | LULC analysis and prediction |
| Digital elevation model (DEM) | Open Topography (Copernicus 30) | Model DEM input |
| Population Density | Population and Housing Census 2011 | LULC Prediction |
| Road Network | Google Earth Pro | LULC Prediction |
| Canal Cross-Section (April 2022) | Bathymetry Survey | Model input |

This drainage network is shown in Figure 8. which illustrates sub-catchment boundaries, the layout of the natural canal (surveyed cross-sections) as well the primary artificial conduits within the study area. The model consists of 76 sub-catchments, which represent different areas of land that contribute to the overall drainage system. It has a total of 374 junctions, which represent the points in the system where the flow converges and diverges. It also includes one outfall at the Goranchatbari pumphouse (Figure 8), opposite the Turag River. A pump element is used to simulate the effect of the pump station on the drainage system. It simulates the effect of pumping water from lower areas within the embanked Goranchatbari catchment to the Turag River to maintain water levels below 3.5 m.

Land use parameters, such as imperviousness and storage capacity, are calculated from the LULC analysis. These parameters are used to estimate the amount of runoff generated by different land uses and to model the behavior of storage facilities, such as ponds and tanks. Calibration parameters, such as Manning's roughness coefficient ($n$) and storage coefficients, are used to adjust the model to match the observed behavior of the drainage system. The n value was taken as 0.04 for the canal bed and 0.06 for the banks. This indicates that the canal bed had a smoother surface than the banks, which would result in less resistance to flow in the channel. The higher coefficient for the banks suggests that the banks were rougher, which would increase the resistance to flow. This was considered to take into account the effect of significant pollution and waste disposal in the canal system [64]. The pollutants can build up on the channel bed and banks, making them more resistant to flow. A list of input parameters for the PCSWMM model is given in Table 4. In addition to these parameters, the study also reported flow routing and continuity errors below 1%. These errors were within acceptable limits and suggest that the model was able to accurately simulate the flow of water through the drainage system [65].

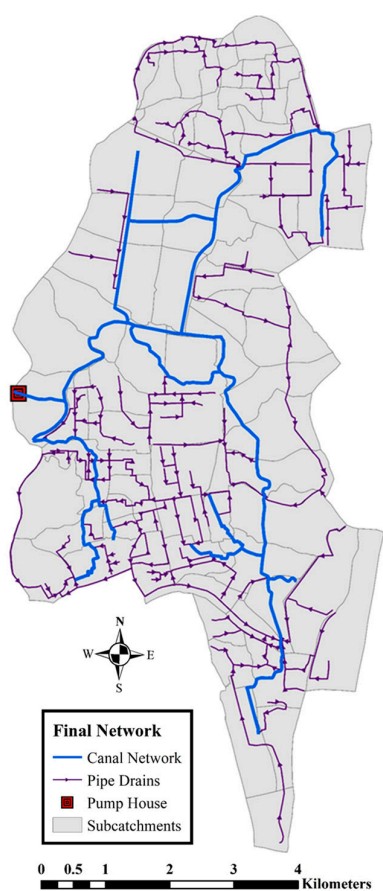

**Figure 8.** PCSWMM model: Goranchatbari drainage network (natural canals and artificial conduits) and sub-catchment boundary.

**Table 4.** PCSWMM model parameters and methods.

| Parameter | Values |
|---|---|
| Junction baseline flow | 0.002 m$^3$/s |
| Sub-catchment slope | 0.002% |
| Infiltration method | Green-Ampt |
| Suction head | 3.5 mm |
| Soil conductivity | 0.5 mm/h |
| Routing method | Dynamic Wave |
| Force equation | Hazen-Williams |
| Surcharge method | Extran |

*2.4. Boundary Condition*

The rainfall frequency analysis was done to calculate the total precipitation for 2.33-, 25-, and 50-years return period to use as the boundary conditions for three different cases. The yearly maximum precipitation between 1965–2020 was fed into the Gumbel distribution [66] to generate rainfalls of 2.33-, 25-, and 50-years of the return period. The 24-h rainfall hydrographs (Figure 9) were generated based on the [67] reports for Dhaka's stormwater improvement project. The precipitation attained its maximum intensity of 149.28 mm for the 50-year return period, while for the 25-year return period, this value was recorded as 118.04 mm. Similarly, for the 2.33-year return period, the precipitation peak was found to be 59.66 mm.

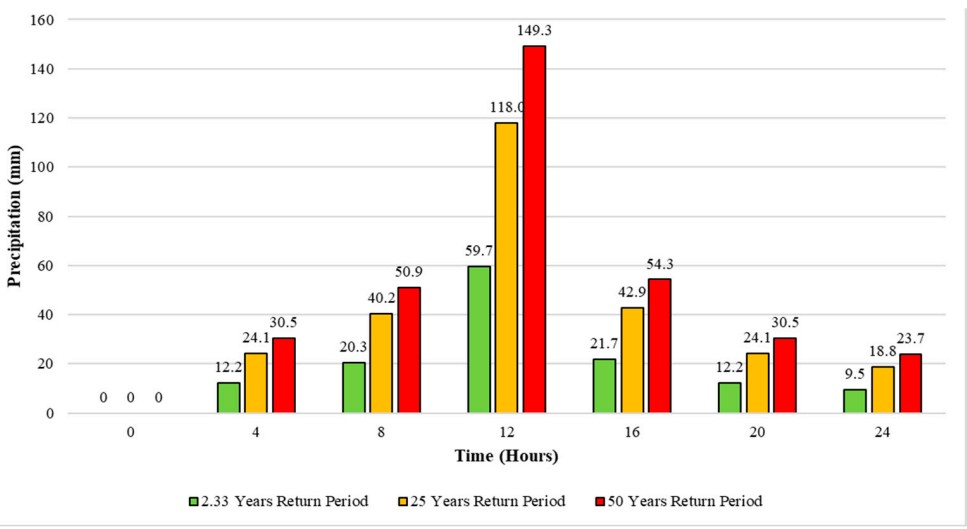

**Figure 9.** Rainfall hyetograph for 2.33, 25, and 50 Years of Return Period.

## 3. Results and Discussions

### 3.1. Validation Results

The GPS gauge was the only source of observed water level data to validate the drainage model. The validation period covered 25 days, from 4 May to 28 May 2020. This timeframe allowed us to analyze a series of events: initially, there was sporadic rainfall from 6 May 2020, to 18 May 2020, followed by two significant rainfall events on 20 May and 24 May 2020. We chose this validation window to accurately represent the water level at the GC pump house throughout this entire period. It's important to note that our choice of this timeframe was also influenced by the availability of reliable water level records at GPS.

GPS has a total of 8 fully functional pumps, each capable of draining out water at a flow rate of 8 m$^3$/s, giving the pump station a total capacity of 66 m$^3$/s. However, at any given time, only one pump is kept operational. According to the official statement, the pump is turned on when the water level reaches 3.5 m. To reflect this in the PCSWMM model, the pump element was set to have a capacity of 8 m$^3$/s, with a conditional operational clause that would turn on the pump as soon as the water level rose above 3.5 m. However, during the month of May 2020, the pump operation log revealed that the pump operator did not always turn on and off the pump at the set threshold of 3.5 m. Instead, the pump was turned on when the water level reached a range between 3.25 and 3.75 m. Additionally, there were times when more than one pump was operational for a short period of time in order to reduce the water level rapidly during heavy precipitation events. This variability in pump operations cannot be simulated by the pump element incorporated in the PCSWMM model, and as a result, a constant draw out of 8 m$^3$/s was assumed for the entire model validation period. Even with these practical limitations, the model responds to the sudden influx of rainwater during high precipitation events. In addition, for the simulation period from 4–28 May 2020, Figure 10a shows the correlation coefficient of the model simulated and observed data to be 0.84. Figure 10b shows the sub-catchment runoff coefficient with high accuracy having an R$^2$ value of 0.99. This analysis was done by manually calculating the runoff coefficients of different sub-catchments using land use data and Table 5. The percentage of different land type was the driving factor in calculating this and later comparing it with the model simulated ones [68,69].

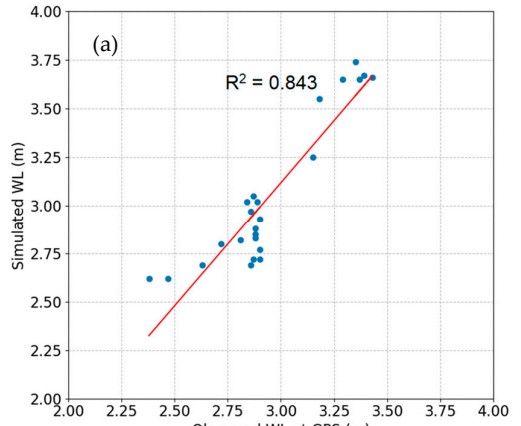
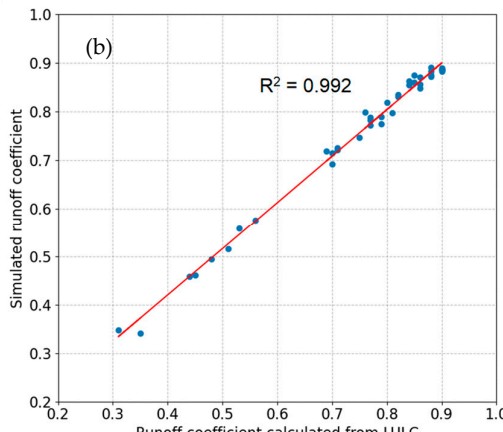

**Figure 10.** Model Validation: (**a**) Correlation between observed and simulated water level (24-h interval) at GPS; (**b**) Correlation between observed and simulated runoff coefficient over the Goranchatbari catchment.

**Table 5.** Runoff Coefficient Chart by Ohio Department of Transportation Hydraulics Manual.

| Runoff Coefficients for Rational Method | | | |
|---|---|---|---|
| | **Flat** | **Rolling** | **Hilly** |
| Pavement & Roofs | 0.90 | 0.90 | 0.90 |
| Earth Shoulders | 0.50 | 0.50 | 0.50 |
| Drivers & Walks | 0.75 | 0.80 | 0.85 |
| Gravel Pavement | 0.85 | 0.85 | 0.85 |
| City Business Areas | 0.80 | 0.85 | 0.85 |
| Apartment Dwelling Areas | 0.50 | 0.60 | 0.70 |
| Light Residential: 1 to 3 units/acre | 0.35 | 0.40 | 0.45 |
| Normal Residential: 3 to 6 units/acre | 0.50 | 0.55 | 0.60 |
| Dense Residential: 6 to 15 units/acre | 0.70 | 0.75 | 0.80 |
| Lawns | 0.17 | 0.22 | 0.35 |
| Grass Shoulders | 0.25 | 0.25 | 0.25 |
| Side Slopes, Earth | 0.60 | 0.60 | 0.60 |
| Side Slopes, Turf | 0.30 | 0.30 | 0.30 |
| Median Areas, Turf | 0.25 | 0.30 | 0.30 |
| Cultivated Land, Clay & Loam | 0.50 | 0.55 | 0.60 |
| Cultivated Land, Sand & Gravel | 0.25 | 0.30 | 0.35 |
| Industrial Areas, Light | 0.50 | 0.70 | 0.80 |
| Industrial Areas, Heavy | 0.60 | 0.80 | 0.90 |
| Parks & Cemeteries | 0.10 | 0.15 | 0.25 |
| Playgrounds | 0.20 | 0.25 | 0.30 |
| Woodland & Forests | 0.10 | 0.15 | 0.20 |
| Meadows & Pasture Land | 0.25 | 0.30 | 0.35 |
| Unimproved Areas | 0.10 | 0.20 | 0.30 |
| Rolling = ground slope between 2% to 10% | | | |
| Hilly = ground slope greater than 10% | | | |

Table 6 shows the values of Percent bias (PBIAS), Nash-Sutcliffe efficiency (NSE), and RMSE-observations standard deviation ratio (RSR), and all of these scores fell into the category of satisfactory range. Therefore, the results indicate that the model is reliable and can be used to study the extreme scenario response and urban flooding.

**Table 6.** Statistical goodness-of-fit test scores of numerical model results.

| Statistical Parameter | WL Validation | Runoff Coeff. Validation |
|---|---|---|
| PBIAS (%) | 3.494 | 1.573 |
| NSE | 0.748 | 0.988 |
| RSR | 0.491 | 0.110 |

*3.2. LULC Change*

Figure 11 depicts the Goranchatbari catchment's classified LULC maps (1973–2022), which are divided into three groups: waterbodies and canals, vegetation, and settlements, and bare lands. Figure 11a–c depicts the spatial patterns of LULC changes in 1973, 1980, and 1993. Natural vegetation, as well as canal and water bodies, were the dominant land use types during this period, while the trajectory of urban expansion (settlement and bare land) shows a southward progression toward Pallabi and Kafrul. Figure 11c,d depicts how, between 1993 and 2003, the built-up or settlement area replaced the majority of the water bodies, canals, and low-lying depressions, as well as the natural vegetation. This resulted in the depletion of wetland areas, which serve as ecological buffers and runoff sinks during heavy precipitation events. The north-eastern settlement region experienced significant expansion, primarily at the expense of vegetative land. The analysis of LULC changes in the Goranchatbari catchment from 2003 to 2013 (Figure 11d,e) shows that the settlement area increased by 23.5 km$^2$ or more than 2.3 km$^2$/year on average. Prior to this rapid increase in settlement, urban growth averaged 0.19 km$^2$/year from 1973 to 1993. In fact, between 2003 and 2013, the area of vegetation decreased by 60 percent, from 25.7 km$^2$ (19.24 percent) to 10.6 km$^2$. Similarly, the area of waterbodies and canals fell to 9.8 km$^2$ (17.79 percent) in 2013 from an already low 17.7 km$^2$ (32.12 percent) in 2003.

According to the 2003 LULC analysis (Figure 11d), the areas around Uttara Khal in the northeast and the Eastern Housing Khal in the lower southwest experienced a significant increase in settlement growth between 1993 and 2003. Despite this rapid settlement growth, Figure 11d shows that in 2003, a significant proportion of the drainage basin (78.77 percent) was still a central waterbody, depressed regions, and vegetation. The primary drainage routes of the northern and south-central regions are found to be Diabari Khal and Buania Khal, respectively. However, Figure 11e shows that the Baunia Khal's extent decreased significantly between 2003 and 2013, with a significant portion of it becoming an urban settlement by 2022. According to [34], Dhaka experienced a 40.17 percent decline in wetlands and waterbody between 1978 and 2009. Within a comparable time range of 1980–2013, a reduction rate of 22.3 percent is found in this study. This suggests that the level of intrusion into aquatic wetlands and waterbodies within the Goranchatbari catchment is lower than the overall patterns observed throughout Dhaka. In fact, Figure 11d,e the GPS retention pond is the only natural water body that has been preserved between 2003 and 2022. However, Landsat images (Figure 11f) show that between 2013 and 2022, approximately 50 acres of land on the northern side of the retention basin were converted to settlement. Over a 50-year period (Figure 12), the water body and vegetation decreased to 15.58 percent and 6.88 percent, respectively, from 43.19 percent and 44.83 percent. Settlements and bare lands, on the other hand, increased by 65 percent in five decades to 77.54 percent from 11.98 percent. Notably, Ref. [22] reported a 42 percent increase in Dhaka's built-up area between 1978 and 2018. It should be noted that the reported rate of increase is lower than that observed in the Goranchatbari catchment area. This strongly suggests that our study area is experiencing an even faster rate of urbanization, surpassing the already rapid rate of urban development within Dhaka city itself.

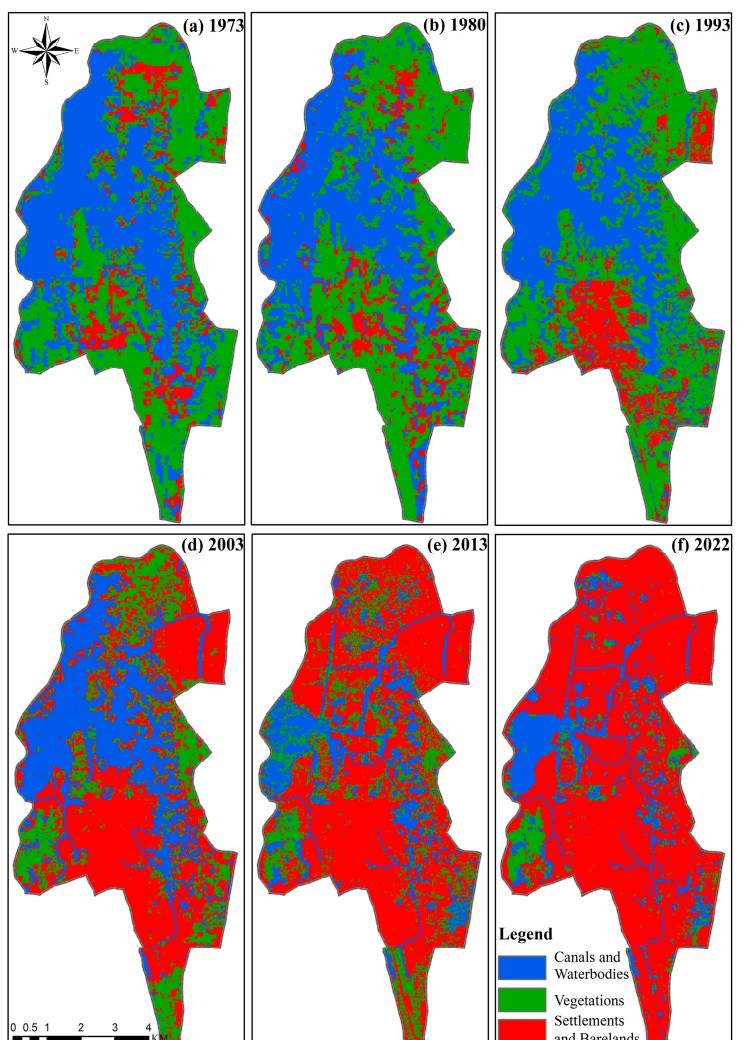

**Figure 11.** LULC change for (**a**) 1973; (**b**) 1980; (**c**) 1993; (**d**) 2003; (**e**) 2013; (**f**) 2022—classified into 3 subgroups—canal and waterbodies, vegetations, settlement, and bare lands.

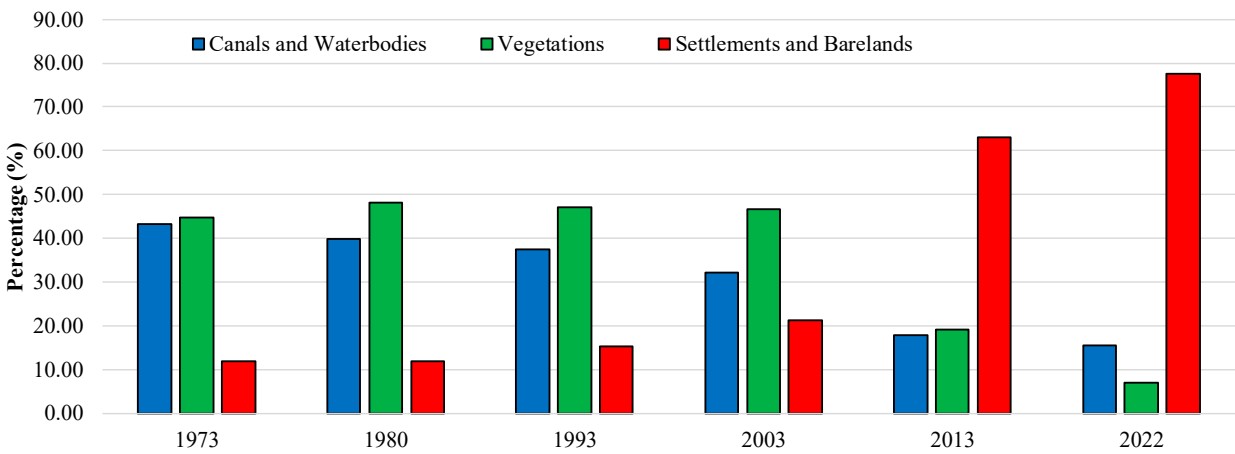

**Figure 12.** LULC change percentage for 3 subgroups—canal and waterbodies, vegetations, settlement, and bare lands.

Illegal encroachment on waterways and ponds has clearly reduced the amount of open space available for water to flow and be stored. Furthermore, the rate of urban development, population influx, and drainage technology adaptation determine how the

Goranchatbari catchment's future pluvial flood conditions will be. Figure 13 depicts the predicted LULC in 2042 to help understand these implications. According to the graph, the settlement area will increase from 77.54 percent in 2022 to 89.49 percent in 2042. This upward trend implies that a greater proportion of land will be used for residential and commercial purposes, potentially depleting natural waterbodies and green spaces. The map also highlights the significant depletion of low-lying terrain suitable for urban settlements, particularly in the vicinity of Baunia Khal. Furthermore, the map indicates that Diabari Khal and Abdullahpur Khal in the northwest region are expected to lose connectivity with Uttara Khal, potentially causing drainage congestion in the northern region. Mirpur Housing Khal and Shangbadik Colony Khal, which collect rainwater runoff from Pallabi and Kafrul, would also be disconnected from the shrinking Baunia Khal. This could exacerbate flooding and other water-related problems in the southeast. The proposed LULC also implies that the Goranchatbari retention pond will not be directly linked to the southwest region's Rupnagar Khal and Bayesteki Khal. Furthermore, the western portion of the Goranchatbari retention pond, adjacent to the embankment, is expected to lose 21.35 percent of its retention area by 2042. Given that the retention pond serves as a reservoir for water before it is discharged into the Turag River, such encroachment has the potential to exacerbate the urban flood vulnerability.

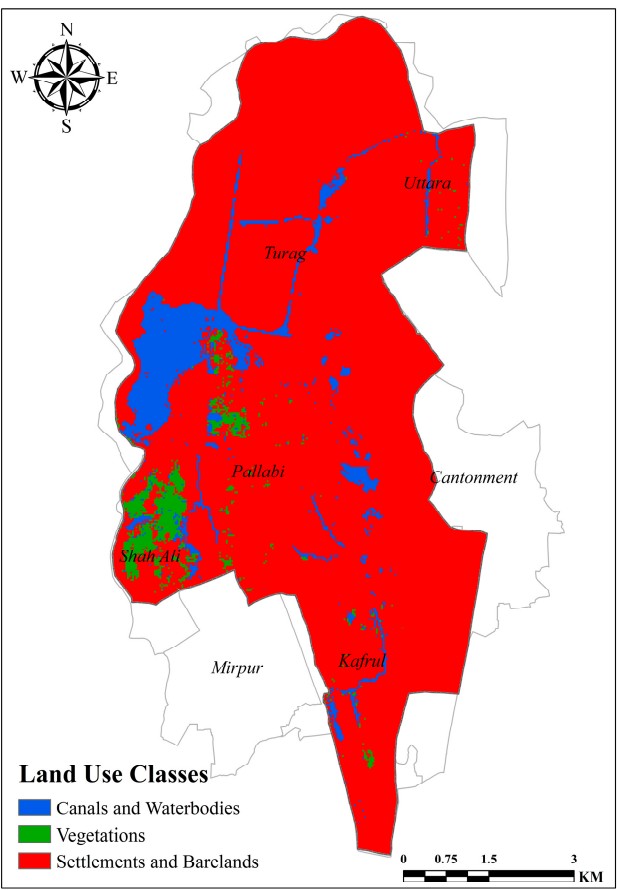

**Figure 13.** Predicted LULC classification for 2042 using MOLUSCE.

### 3.3. Goranchatbari Flooding from Heavy Rain: Present and Future

The hydraulic model is applied to rainfall events with return periods of 2.33-, 25-, and 50-years in both current and future scenarios to investigate the potential impact of rapid urbanization on urban flooding. The first three scenarios (Figure 14a–c) are used to assess the capacity of the existing drainage system in the Goranchatbari area to handle extreme rainfall events. These scenarios are denoted as EH 2.33 RP, EH 25 RP, and EH 50 RP, based on the existing drainage network and historical rainfall. The second three

scenario runs (Figure 14d–f) are designed to assess the future flooding condition in the Goranchatbari area as a result of rapid urbanization. The drainage network is updated using the CA-ANN-based MOLUSCE plugin in QGIS based on the projected LULC maps for 2042. These scenarios are denoted as PH 2.33 RP, PH 25 RP, and PH 50 RP, based on the projected drainage network and historical rainfall.

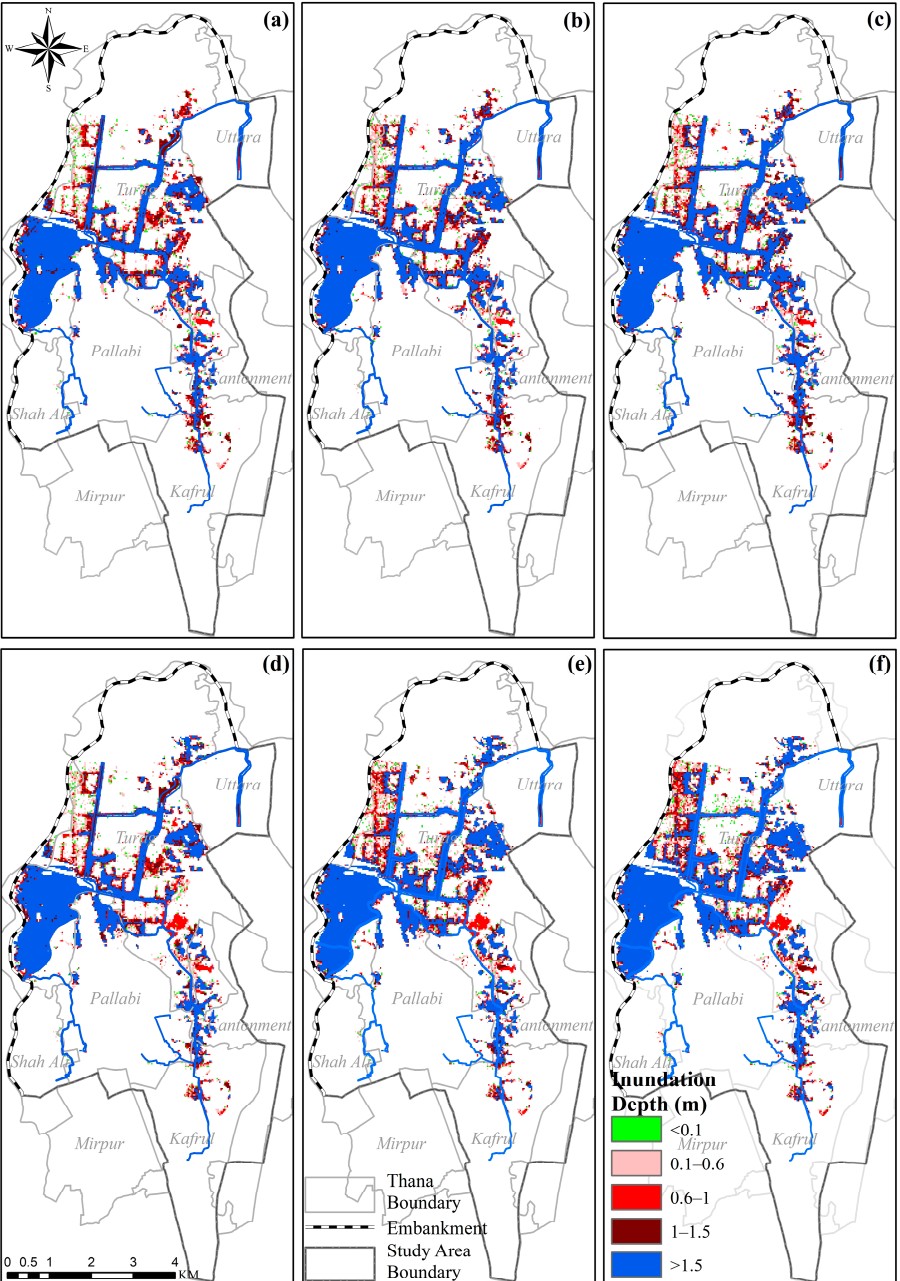

**Figure 14.** Maximum flood inundation depth under (**a**) EH_2.33 RP, (**b**) EH_25 RP, (**c**) EH_50 RP, (**d**) PH_2.33 RP, (**e**) PH_25 RP, (**f**) PH_50 RP within the Goranchatbari Catchment Maps Scenario conditions.

Figure 14 depicts the flood inundation area, which represents the depth and spatial extent of flood waters. When the water depth is less than 0.1 m, the area is waterlogged, but the water only reaches ankle height. The urban area is considered vulnerable if the water depth ranges from 0.1 m to 0.6 m, with water reaching up to knee height. The urban area is classified as severely waterlogged when the water depth ranges from 0.6 m to 1.0 m

and the water level is around waist height. If the water depth is greater than 1.0 m, the area is classified as low-lying lands, khals, natural drainage canals, and retention ponds within the sub-catchment. The classification was done based on the response of field surveys carried out to assess the extent of waterlogging severity within the study area. Residents conveyed the depth of flooding by using relatable terms like "ankle-deep", "waist-deep", "knee-deep", and so on. Consequently, the classification in this paper was designed to align with these field observations for consistency.

According to the EH flood inundation analysis (Figure 14a–c), the west side of the Diabari khal, which is located under Turag thana, is prone to repeated waterlogging in several return period floods. The areas near the Baunia khal that carry stormwater runoff from Mirpur, Pallabi, and Kafrul are vulnerable to high water levels under 2.33, 25, and 50-year rainfall events. The EH 2.33 RP inundation map in Figure 14a shows an inundation depth of 1.5–2.0 m around Baunia Khal. However, as the connectivity is disrupted in the projected LULC scenario: PH 2.33 RP (Figure 14d), the inundation depth and spatial extent increase in the upstream region. Similar characteristics are observed in the western part of the Diabari khal, where the extended flood inundation area increases by 8.61 percent under PH 2.33 RP.

The inundated area around Abdullahpur Khal, which serves as the primary drainage channel for Uttara and Turag thanas, sees a significant increase in water depth in the low-lying floodplain zone under EH 25 RP (Figure 14b). The inundation map shows that due to the LULC change in 2042, a 0.23 km$^2$ area near the intersection of Abdullahpur and Buania khals shifts from a water level below 2 m (Figure 14b) to a water level above 2 m (Figure 14e). Because of increased urbanization and encroachment in the area, this change in water depth is attributed to a decrease in connectivity and disruption in the natural drainage system. Similarly, the downstream portion of the Baunia khal exhibits a shift in water depth and spatial inundation extent toward greater depth. These findings are consistent with recent research by [31], which shows a high vulnerability to waterlogging around the Baunia Khal depression.

The EH 50 RP (Figure 14c) and PH 50 RP (Figure 14f) runs show the inundation extent to be more severe around the west portion of Turag thana, Diabari Khal. This area is naturally low-lying and depressed, and it is bounded on the west side by the Turag River embankment, making it more vulnerable to flooding. The Diabari Khal's lack of drainage connectivity exacerbates the problem, as heavy rainfall causes runoff to accumulate faster, resulting in severe waterlogged conditions. The upper portion of the Abdullahpur Khal is seen to have increased water depth due to channel width shrinkage, which is primarily attributed to the LULC change and increase in urban settlements in PH 50 RP. As cities grow and encroach on the natural environment, natural drainage systems are disrupted, resulting in increased flood inundation.

Figure 15 depicts a comparison of waterlogged areas with a 1 m inundation depth under current (EH) and projected (PH) LULC conditions for 2.33, 25, and 50 years of return period runs. The depth of inundation is divided into five categories, each with a 0.2 m increment from 0 to 1 m. The results show that the inundation area between 0.4–0.8 m of water depth remains nearly the same in both current and projected LULC conditions, but the PH 50 year has a 31% increase in inundation area over 0.8 m compared to EH 50-year. In the 25-year return period simulation (Figure 15b), PH 25 year increases by 9.52 percent and 19.05 percent in the 0.6–0.8 m and 0.8–1.0 m ranges, respectively. The PH 25 year average increase for inundation areas below 0.6 m water depth is 5.18 percent. Figure 15c shows a significant increase in inundation levels between 0.6 and 0.8 m, indicating a 21.34 percent increase over baseline conditions. According to the analysis, the total inundation area below the 1.0 m water depth threshold increases by 8.47 percent, 8.11 percent, and 4.05 percent in PH cases for 50-, 25-, and 2.33-year return period rainfall events, respectively, compared to the current conditions in 2022. This means that more land is at risk of being inundated during heavy rainfall events, potentially resulting in increased flood damage and asset loss.

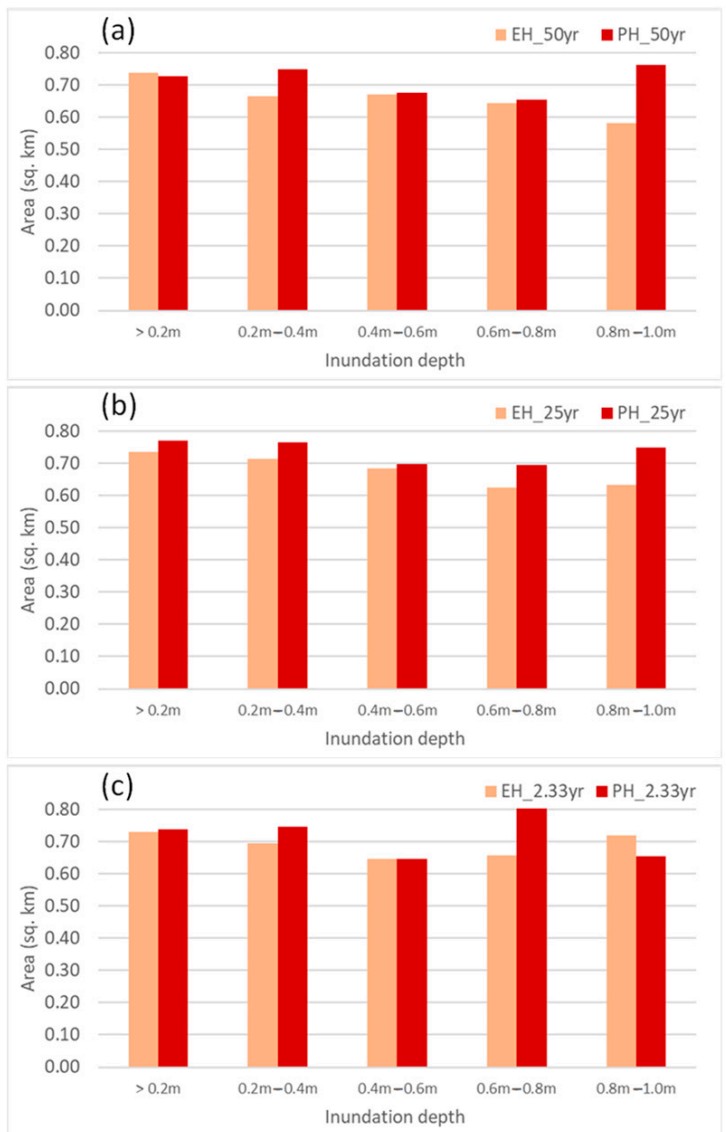

**Figure 15.** Inundated area under (**a**) 50 years (**b**) 25 years (**c**) 2.33 years return period precipitation within the Goranchatbari catchment.

Another special projected scenario (SPH) has also been simulated to assess the impact of extreme events in 2042 if minimum amount of connectivity among the dead spots of the canals are maintained (1 m width). It has been found that, the total inundation extent does not quite vary with the PH scenario (Figure 16). However, the depth of the inundation varies to some extent. For, 50 years return period, the area of the zone in which the inundation layer is above 0.8 m will decrease by a small margin of 2.32%. The rest of the analysis for this return period remained unchanged. Scenarios like, SPH 2.33 year, and SPH 25 year faces negligible amount of change comparing to PH 2.33 year, and PH 25 year return period scenarios respectively. This proves the fact that, ensuring a minimum amount of connectivity will not bring much positives on a larger scale. So, to ensure a sustainable solution, the canals need to be freed from every sort of encroachments.

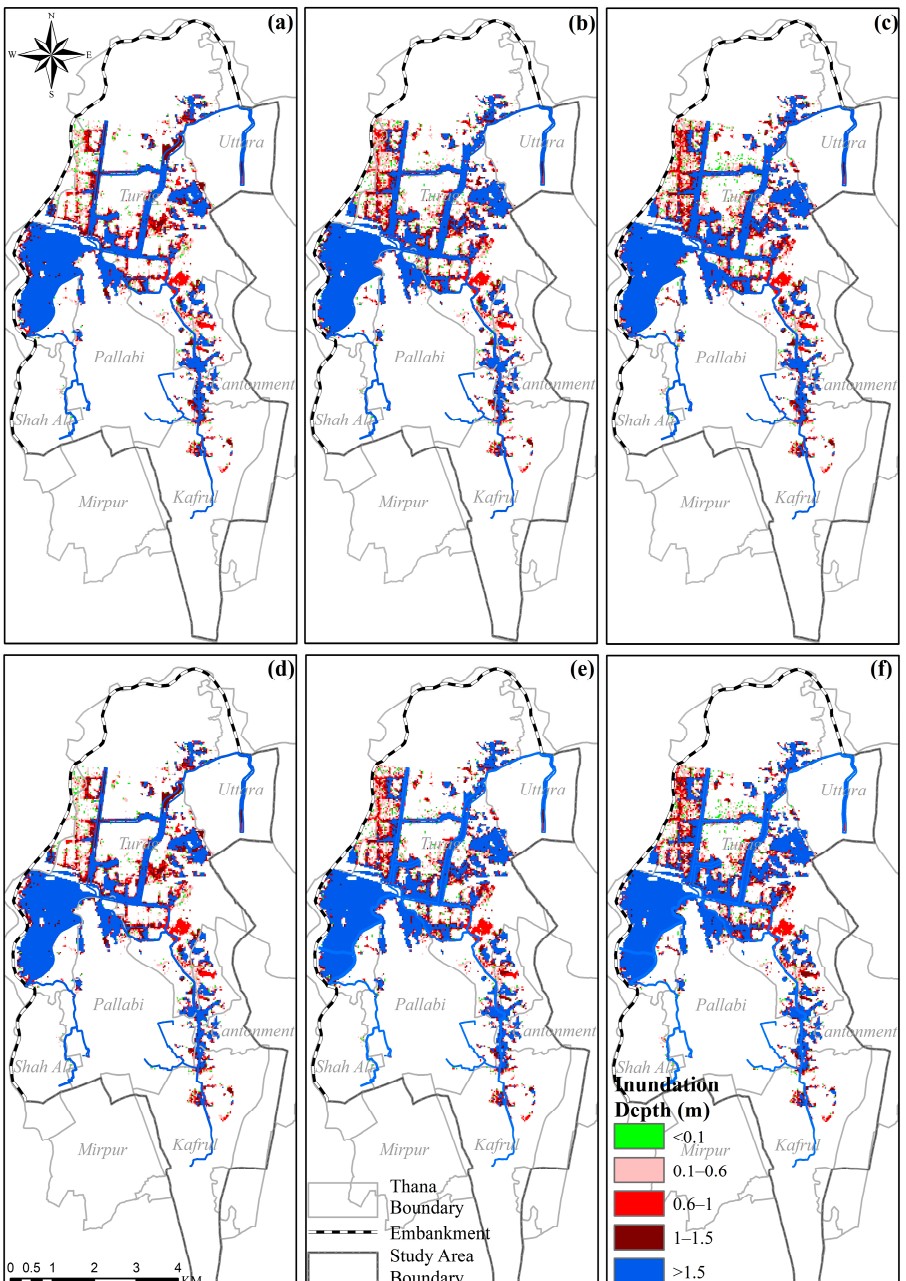

**Figure 16.** Maximum flood inundation depth under (**a**) SPH_2.33 RP, (**b**) SPH_25 RP, (**c**) SPH_50 RP, (**d**) PH_2.33 RP, (**e**) PH_25 RP, (**f**) PH_50 RP within the Goranchatbari Catchment Maps Scenario Conditions.

Table 7 and Figure 17 show the duration of flooding caused by different return period rain patterns in existing (EH) and altered (PH) land use. When these cases are compared, the no-flood area is reduced by 3% to 10% as a result of the altered land use. In the case of EH cases, the high duration (2 to 5 days) was observed in nearly 4 to 8% of the catchment areas, increasing to 8 to 12% in altered land use conditions. Due to altered land use conditions, an additional 15% of areas would like to experience 1 h to 5 days of flooding. Figure 17 indicates that Baunia Khal, Parise Road Khal, Shangbadik Colony Khal, Mirpur Cantonment Khal, Abdullahpur Khal, and Digun Khal will face devastating floods in 2042 that will likely last 2–5 days.

**Table 7.** Duration of the flooding.

| Duration of Flooding | Percentage of Inundation | | | | | |
|---|---|---|---|---|---|---|
| | EH | | | PH | | |
| | EH_2.33 | EH_25 | EH_50 | PH_2.33 | PH_25 | PH_50 |
| No Flooding | 55 | 50 | 43 | 48 | 40 | 40 |
| <30 min | 29 | 32 | 39 | 31 | 38 | 38 |
| 1–6 h | 1 | 1 | 0 | 2 | 1 | 1 |
| 6–24 h | 5 | 3 | 4 | 5 | 3 | 3 |
| 1–2 Days | 6 | 5 | 5 | 6 | 7 | 6 |
| 2–5 Days | 4 | 8 | 8 | 8 | 11 | 12 |

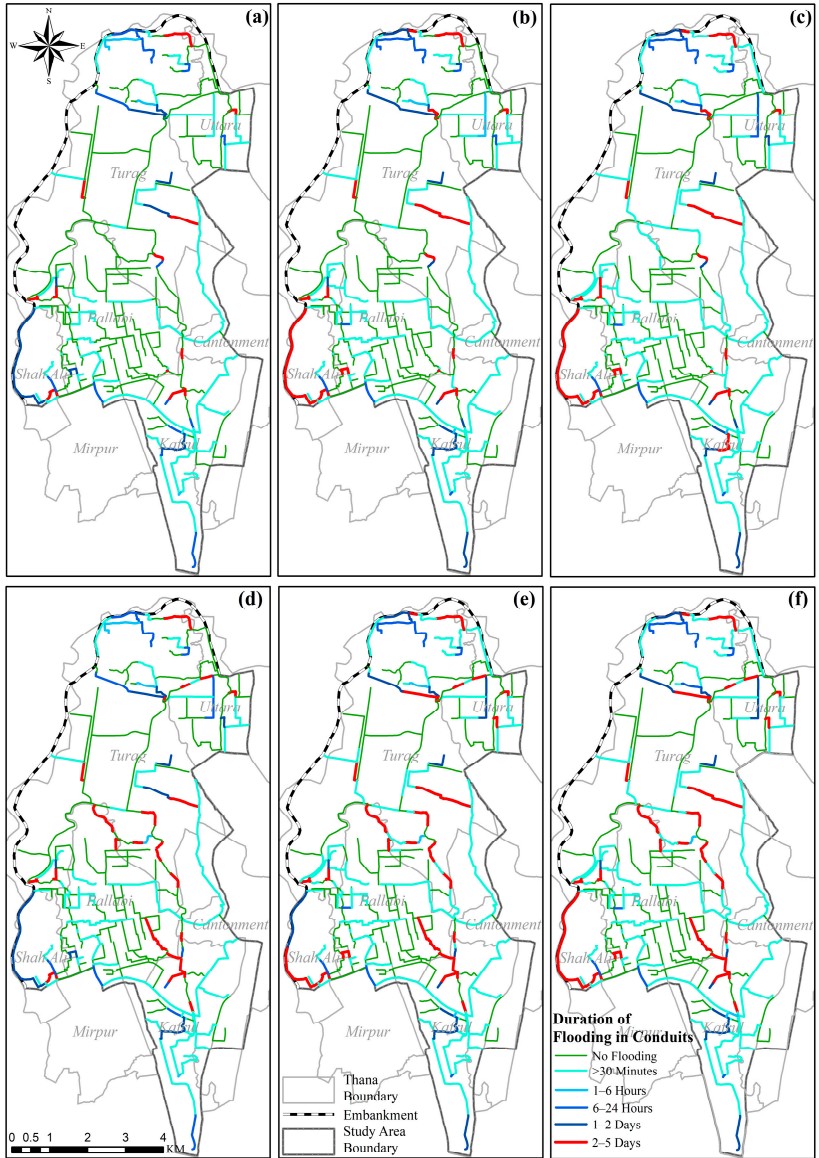

**Figure 17.** Sub-catchment-wise Flood Duration Map (**a**) EH_2.33 RP, (**b**) EH_25 RP, (**c**) EH_50 RP, (**d**) PH_2.33 RP, (**e**) PH_25 RP, (**f**) PH_50 RP within the Goranchatbari Catchment Maps Scenario conditions.

## 4. Conclusions

The Goranchatbari catchment is one the last remaining interconnected natural drainage systems of Dhaka city. The rapid pace of urban development has put significant demo-

graphic pressure on an already fragile system. Field observations revealed that unchecked urban development and poor waste management practices were crippling the natural drainage system of the catchment. Furthermore, the construction of residential infrastructure over already encroached khals and low-lying areas, as well as dumping of unchecked solid waste and industrial effluent, are the primary factors reducing the conveyance capacity of the khals. According to LULC analysis, settlement/built-up area has increased by more than 2.3 km$^2$/year on average between 2003 and 2013. This rate of urban growth during 2003–2013 is 12 times of the growth between 1973 and 1993. The western portion of Turag thana is highly susceptible to urban flooding due to its naturally depressed topography and absence of a well-integrated drainage system with the primary khal network. Low-lying areas near the Baunia Khal and the upstream region of the Abdullahpur Khal will be highly vulnerable to urban flooding in the future. For 50 years, 25 years, and 2.33 years of rainfall events, the current trajectory of LULC change results in an increase of 8.47%, 8.11%, and 4.05% in the total inundated areas (<1.0 m) in 2042, respectively, in comparison to the current condition. Flood duration is also expected to increase as a result of LULC change. Long-term flooding will result in 11% more areas.

The findings of the study can assist policymakers and city planners in understanding the relationship between urban flooding and unchecked urban growth. Stakeholders can take proactive steps to mitigate future flooding risks and enhance long-term sustainability and resilience of the city by assessing the potential risks of future waterlogging.

**Supplementary Materials:** The following supporting information can be downloaded at: https://www.mdpi.com/article/10.3390/w15213834/s1. Section S1: Status of Natural Canals or Khals in the Study Area.

**Author Contributions:** Conceptualization, S., M.S.M. and M.S.S.; methodology, S., M.S.S. and M.S.M.; software, S.A. and M.S.S.; validation, S.A. and M.S.S.; formal analysis, S.A., M.S.S., S., S.B.M. and R.K.; investigation, S.A., M.S.S., S., M.S.M., S.B.M., R.K. and A.I.A.C.; resources, M.S.M.; data curation, S.A., M.S.S. and R.K.; writing—original draft preparation, M.S.S., S.A. and S.; writing—review and editing, M.S.S., S.A., S. and M.S.M.; visualization, S.A., M.S.S., S. and M.S.M.; supervision, M.S.M.; project administration, M.S.M.; funding acquisition, M.S.M. All authors have read and agreed to the published version of the manuscript.

**Funding:** This research was funded by the U.S. Department of State under the project "Assessing Health Impacts of Urban Flooding under Changing Climate: A Case Study in Dhaka City".

**Data Availability Statement:** Data is available on request.

**Acknowledgments:** The authors acknowledged the contributions of different key stakeholders who provided valuable information during the field survey.

**Conflicts of Interest:** The authors declare no conflict of interest.

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
