# Peer review of "Impact of Urbanization on Pluvial Flooding: Insights from a Fast Growing Megacity, Dhaka"

_water, doi:10.3390/w15213834_

Round 1

Reviewer 1 Report

Comments and Suggestions for Authors

A separate file has been attached. I have also marked some suggestions in the manuscript.

Review comments

The manuscript entitled “Impact of urbanization on pluvial flooding: Insights from the fastest of growing megacity in the world, Dhaka” by Sakib and the co-authors is found to be suitable for the journal. The manuscript deals with serious concern on LULC changes and consequently flooding pattern in one part of Dhaka. I am suggesting some corrections as given below.

1.     Flood frequency analysis - Why 2.33-year return period is chosen for the analysis? I can understand 25- and 50-year but 2.33-year return period seems to be odd.

2.     All the mathematical variables in the text should be written in italics.

3.     The LULC prediction model for 2042 has been run without giving any constraints. I would like to see the prediction with one more scenario – the minimum hydraulic connectivity among the water bodies and canals will be maintained in future. Under this circumstance, predict fluvial flooding pattern and analyze. This is required because authority may try to maintain, at least, the dewatering process.

4.     Please mention how pumps are operated in the future scenario. Do you suggest additional pumping arrangements for the future scenario after maintaining minimum hydraulic connectivity among the waterbodies. This step will ensure some kind of solution measure instead of just highlighting the problems.

5.     The optional suggestion – The return period rainfalls are considered for future scenario with LULC in 2042. In my opinion, the authors should consider climate projected rainfall over the region. The CORDEX data can be analyzed and checked for change in rainfall pattern. The rainfall during the observed flood event can be modified considering the percentage increase/decrease in rainfall due to climate change. The scenario can be projected as the change in flooding pattern if the waterbodies and open areas are encroached. The civic society and the city authority can better apprehend the problem. The authors may follow this article:

Nithila Devi, N., Sridharan, B., Kuiry, S. N. (2019). Impact of urban sprawl on future flooding in Chennai city, India. Journal of Hydrology, 574, 486-496.

6.     Additional suggestions are marked in the attached file.

Author Response

Thank you for your positive comments. We have tried to address all of your comments.

Reviewer 2 Report

Comments and Suggestions for Authors

See comments in the attached file. 

Comments on the Quality of English Language

The paper is overall understandable, but there are some confusing sentences. Grammar needs to be checked.

Author Response

Thank you very much for your detailed comments. We tried to address all of those

Round 2

Reviewer 2 Report

Comments and Suggestions for Authors

Thanks for the detailed responses and for clarifying some passages.

The 1D flood model still needs better clarification.

-It's ambiguous whether additional cross-sections were incorporated between the 126 that were surveyed. If the new cross-sections have been included, the transition between them, especially in close proximity to the surveyed cross-sections, needs explanation due to the lack of interpolation.

-Were floodplains modeled both upstream and downstream relative to the most distant cross-sections in either direction? In that case, please describe how cross section were extrapolated?

-What is the average spacing between cross sections of the final 1D model?

-I suggest adding a figure of your 1D model and indicating the cross sections that were surveyed.

Please also be sure to include the modifications described in the response file into the paper. I noted that figure 9 of the modified manuscript, for instance, still shows hydrograph despite the authors saying the the manuscripts has been updated.
